# W-Band FMCW MIMO System for 3-D Imaging Based on Sparse Array

**Wenyuan Shao** [1,2,3], **Jianmin Hu** [1,4,5,*], **Yicai Ji** [1,2,3], **Wenrui Zhang** [1,2,3] and **Guangyou Fang** [1,2,4,5,*]

1   Aerospace Information Research Institute, Chinese Academy of Sciences, Beijing 100094, China; shaowenyuan18@mails.ucas.ac.cn (W.S.); ycji@mail.ie.ac.cn (Y.J.); zhangwenrui19@mails.ucas.ac.cn (W.Z.)
2   Key Laboratory of Electromagnetic Radiation and Sensing Technology, Chinese Academy of Sciences, Beijing 100190, China
3   School of Electronic, Electrical, and Communication Engineering, University of Chinese Academy of Sciences, Beijing 100049, China
4   GBA Branch of Aerospace Information Research Institute, Chinese Academy of Sciences, Guangzhou 510530, China
5   Guangdong Provincial Key Laboratory of Terahertz Quantum Electromagnetics, Guangzhou 510530, China
*   Correspondence: hujm@aircas.ac.cn (J.H.); gyfang@mail.ie.ac.cn (G.F.)

**Abstract:** Multiple-input multiple-output (MIMO) technology is widely used in the field of security imaging. However, existing imaging systems have shortcomings such as numerous array units, high hardware costs, and low imaging resolutions. In this paper, a sparse array-based frequency modulated continuous wave (FMCW) millimeter wave imaging system, operating in the W-band, is presented. In order to reduce the number of transceiver units of the system and lower the hardware cost, a linear sparse array with a periodic structure was designed using the MIMO technique. The system operates at 70~80 GHz, and the high operating frequency band and 10 GHz bandwidth provide good imaging resolution. The system consists of a one-dimensional linear array, a motion control system, and hardware for signal generation and image reconstruction. The channel calibration technique was used to eliminate inherent errors. The system combines mechanical and electrical scanning, and uses FMCW signals to extract distance information. The three-dimensional (3-D) fast imaging algorithm in the wave number domain was utilized to quickly process the detection data. The 3-D imaging of the target in the near-field was obtained, with an imaging resolution of 2 mm. The imaging ability of the system was verified through simulations and experiments.

**Keywords:** multiple-input multiple-output (MIMO); frequency modulated continuous wave (FMCW); sparse linear array; three-dimensional (3-D) imaging; channel calibration

## 1. Introduction

Millimeter waves are high-frequency electromagnetic waves typically defined in the 30~300 GHz frequency band, ranging between microwave and visible light. As a technical means for human security screening, millimeter wave imaging technology has multiple advantages [1–3]. Firstly, millimeter wave systems have good penetrability, so can detect items hidden under clothing or in luggage. Secondly, according to the dielectric properties of different materials, the millimeter wave imaging technique has a good resolution to distinguish different hidden objects and can detect dangerous objects such as plastic guns, ceramic knives, explosives, etc. Thirdly, due to its short wavelength, millimeter wave imaging systems have a high spatial resolution and can accurately obtain the location, size, and shape information of the target through 3-D imaging. Finally, the millimeter wave system adopts a non-contact working mode, with low microwave radiation power and low harm to human health. Millimeter wave imaging has become an important technology for security check imaging. With the development of technology, millimeter wave 3-D security inspection equipment have been widely studied in many countries [4].

Based on the single-input single-output (SISO) synthetic aperture radar (SAR) system, the Pacific Northwest National Laboratory (PNNL) in the United States applied the proposed 3-D holographic imaging algorithm to the column scanning system. In collaboration with the L3 Communication Company, PNNL developed the 'Provision' series of products, which became one of the most mature and best-selling active source security inspection products in the world [5]. However, for imaging scenes with larger target dimensions, the number of transceiver units in the SISO array is still large, which not only increases the system cost but also makes the SISO array cumbersome and complex. It also slows down the scanning speed and affects the overall safety detection time.

In recent years, the application of the MIMO principle in radar enables imaging systems to use fewer array units to achieve greater dynamic range effects compared with the SISO systems [6]. SISO systems have higher efficiency in array utilization, which can further reduce the data acquisition time and hardware cost. MIMO radars are widely used in the active source security screening imaging systems with their low cost and high imaging resolution characteristics.

In 2011, the Jet Propulsion Laboratory (JPL) of the United States developed and processed a set of FMCW imaging radars with operating frequencies of 660~690 GHz for the detection of concealed weapons or dangerous goods at a relatively long distance [7]. The radar can detect targets at the range of $25 \pm 1$ m and generate one $40\,\text{cm} \times 40\,\text{cm}$ image per second in real time. The azimuth resolution could be 1 cm and the range resolution could be 5.2 cm.

In 2011, the German company Rohde & Schwarz collaborated with scholars from the University of Nuremberg to launch the QPS series of security products based on sparse periodic array technology [8–11]. The aperture dimension was expanded to a $2\,\text{m} \times 1\,\text{m}$ rectangular aperture, which includes a total of $4 \times 8$ MIMO antenna clusters, improving the efficiency and quality of security imaging. However, despite sparse optimization of the MIMO array, there are still over 3000 antenna units. High imaging efficiency relies on large-scale computer hardware configurations, which means the QPS systems face high hardware costs.

In 2012, the Institute of Electronics of the Chinese Academy of Sciences (IECAS) proposed and studied a new millimeter wave imaging method for the application of fast imaging detection of illegal objects hidden in the human body [12]. The system adopts a novel millimeter wave imaging method, combining fan-beam scanning and the synthetic aperture principle. The field of view range for imaging is $180\,\text{cm} \times 60\,\text{cm}$. It can complete high-resolution 3-D imaging within 5 s, with an azimuth resolution of 4 mm and a range resolution of 7.8 mm [13]. The system has high imaging accuracy, but it still cannot meet long-distance and high frame rate requirements and is only used for cooperative imaging scenarios.

In 2013, the University of Electronic Science and Technology of China (UESTC) developed an all-solid-state linear FMCW terahertz imaging system, with a center frequency of 0.34 THz and a bandwidth of 7.2 GHz. The imaging method involved inverse synthetic aperture imaging, which realizes centimeter-level azimuth and range resolution [14,15].

In 2015, the Center for Information Processing and Telecommunications of the Universidad Politécnica de Madrid (UPM), Spain, developed a 300 GHz continuous-wave linear frequency modulation (CW-LFM) screening system based on a bifocal ellipsoidal reflector system (BEGRS) [16]. With a field of view of $50\,\text{cm} \times 90\,\text{cm}$ at a standoff distance of 8 m, the system can acquire 3-D images in 0.5 s, with an azimuth resolution of 16 mm and a range resolution of 5.5 mm [17].

In 2017, German scholars Daniela Bleh et al. investigated an FMCW radar imaging system operating in the W-band near 100 GHz [18]. It adopts a geometrically optimized two-dimensional antenna array consisting of 22 transmitters and 22 receivers. The system operates with a bandwidth of no more than 4 GHz and can image targets in the range of 3 m to 60 m.

In 2018, the Institute of Electronic Engineering of the Chinese Academy of Engineering Physics (CAEP) and the National University of Defense Technology (NUDT) jointly carried

out a 340 GHz band, four-transmitter, sixteen-receiver array imaging system. It can achieve an azimuthal resolution of 14 mm and a range resolution of 9 mm at a standoff distance of 4 m [19]. However, the target of the system scenario is stationary, which does not yet meet the requirements for non-cooperative security inspection.

In summary, existing millimeter wave imaging systems have shortcomings such as numerous and complex array units, high hardware costs, narrow bandwidths, and low imaging resolutions. In response to the above issues, this study designed a new 3-D millimeter wave radar imaging system. The system operates at 70~80 GHz, consisting of 22 transmitting units and 160 receiving units. The antenna array consists of ten radio frequency (RF) chips, with the same structure cascaded together, and the periodic array structure improves the utilization rate of the antenna units, thus reducing the hardware costs. The system utilizes a linear sparse MIMO array combined with planar mechanical scanning to form a 2-D aperture. The range information was extracted using FMCW signals. High resolution 3-D imaging results of targets can be quickly achieved by a 3-D imaging algorithm in the wavenumber domain, with an azimuthal resolution of up to 2 mm. In the process of acquiring echo data and data processing, a combined method of separating the 'direct wave' from the original signal and flat-panel correction was adopted to effectively calibrate the channel error of the system and improve the imaging quality.

1.  Existing high-frequency security equipment is mostly used for long-distance security checks; for example, at an imaging distance of 10~50 m, and the azimuth resolution is generally not less than 14 mm. The security equipment used for close-range imaging generally operates at a frequency of 24~35 GHz and has an azimuth resolution of no less than 8 mm. The imaging system proposed in this article operates at a frequency of 70~80 GHz and can achieve a limited azimuth resolution of 2 mm for objects in an imaging area of 10~60 cm, which meets the increasingly improving imaging resolution requirements. This frequency band also balances imaging resolution and electromagnetic wave penetration into objects.

2.  The existing MIMO imaging systems require hundreds or even thousands of transmitting and receiving antennas, even if using sparse array structures. The numerous and complex array units result in high hardware costs. The imaging system proposed in this article adopts a periodic array structure in order to improve the utilization of the antenna units. By combining a one-dimensional sparse MIMO array with mechanical scanning, a two-dimensional planar aperture is formed, which reduces hardware costs.

3.  Conventional channel calibration methods necessitate an additional reference channel positioned separately from the receiving channel [20]. However, these methods have their drawbacks, including an inability to perform real-time calibration, inconvenience in the calibration process, and systematic errors. The self-calibration method in this article utilizes the receiving channel exclusively, and avoids extra hardware. The software separates and calibrates the direct wave, streamlining the process and enabling real-time calibration without electrical parameter drift issues.

This paper is organized as follows. Section 2 describes the processing flow of the MIMO signals and the imaging algorithm, as well as the design scheme of the antenna array. Section 3 describes the design of the radar system, including the system architecture and the system channel calibration method. In Section 4, experiments based on the designed system are conducted and imaging experimental results are presented. Finally, Section 5 summarizes the results of this paper and draws conclusions.

## 2. MIMO Signal Model and Processing

In this section, the signal model and imaging algorithm used for the system are presented, the design principle and scheme of the antenna array are given, and the computer simulations are described.

*2.1. FMCW Signal Model*

In this study, the transmitted waveform of the MIMO radar system was a linear FMCW signal, and the frequency of the signal changed linearly with time. The instantaneous frequency of the transmitted signal $f_T(t)$ can be expressed as

$$f_T(t) = f_0 + \frac{B}{T}t,\ 0 \leq t \leq T \tag{1}$$

where, $B$ is the bandwidth, $T$ is the duration of one cycle of the signal, $f_0 = f_c - B/2$ is the starting frequency of the signal, and $f_c$ is the carrier frequency, which is the center frequency of the transmitted signal. $\mu = B/T$ is the slope of the FM signal [21].

Assuming that the initial phase of the transmit signal is 0, the FMCW radar transmit signal in a single cycle $s_T(t)$ can be expressed in the form of a complex signal as

$$s_T(t) = A \exp\left[-j2\pi(f_0 t + \frac{\mu}{2}t^2)\right] \tag{2}$$

where $A$ is the transmit signal amplitude.

Assuming that the target is stationary, the distance of the target from the transmitting antenna and the receiving antenna are $R_t$ and $R_r$, respectively, and the transmission speed of the electromagnetic wave in the air is $c$. The signal received by the receiving antenna is delayed by $\tau = (R_t + R_r)/c$ compared with that of the transmitting signal. Therefore, ideally, the echo signal $s_R(t)$ of the target received by the receiving antenna can be expressed as

$$s_R(t) = K_r A \exp\left[-j2\pi(f_0(t-\tau) + \frac{\mu}{2}(t-\tau)^2) + \theta_0\right] \tag{3}$$

where $K_r$ is the reflection coefficient of the target and $\theta_0$ is the additional phase caused by the target reflection.

The received echo signal was mixed with the transmitted signal through a mixer, and the high-frequency components in the mixed differential signal were filtered out through a filter. The resulting differential signal is represented as follows

$$s_R(t) = K_r A \exp\left[-j2\pi(f_0(t-\tau) + \frac{\mu}{2}(t-\tau)^2) + \theta_0\right] \tag{4}$$

where $A_0$ is the amplitude of the differential frequency signal. The echo signals obtained above can be used in algorithms for imaging.

The signal model in this paper is applicable to static scenarios and is compatible with most cases of security screening scenarios. However, the proposed radar system can be easily extended to dynamic targets and scenarios by considering Doppler shift and phase correction terms.

*2.2. 3-D MIMO Omega-k Algorithm*

Nowadays, many mature synthetic aperture imaging algorithms have been widely applied to airborne or spaceborne SAR, such as chirp scaling algorithms (CSAs) [22], polar format algorithms (PFAs), range doppler algorithms (RDAs), back projection algorithms (BPs), range migration algorithms (RMAs) [23], etc. In the close-range human security imaging scenario, the target size was comparable to the size of the synthetic aperture; also comparable was the distance from the aperture to the target, and the plane wavefront assumption, also known as the far-field assumption or Fresnel approximation, could not be used. Due to the above limitations, and considering the algorithm's computing speed and computational complexity, RMA was chosen for target imaging in this study. RMA, also known as the frequency–wavenumber domain imaging algorithm, is a method for correcting spherical waveform wavefront bending by interpolating in frequency wavenumber domains. It can efficiently process data with large oblique angles, wide apertures, and large scenes, and is more suitable for close-range human security imaging scenes [24].

A schematic diagram of the radar echo signal geometry model is shown in Figure 1; the 2-D MIMO aperture in the figure is only for schematic illustration. Assuming that the target of the spatial distribution function is $O(x', y', z')$, and its scattering center is specified as the origin of the coordinate system, the $x$-axis indicates the horizontal azimuth, the $y$-axis indicates the vertical azimuth, and the $z$-axis indicates the forward direction of the radar, which is called the range direction.

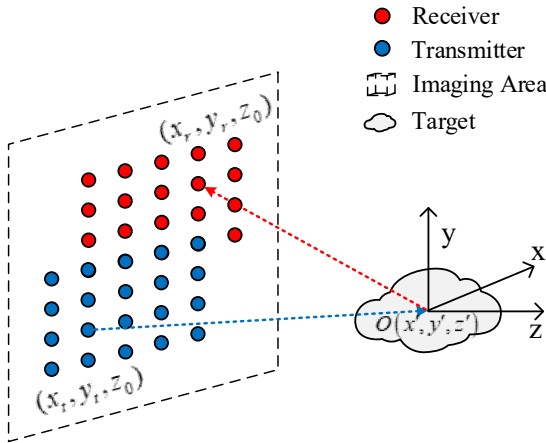

**Figure 1.** Schematic diagram model of radar echo signal.

The phase center plane of the MIMO array is located on the negative axis of the $z$-axis, at a distance of $z_0$ from the origin. The blue dots in Figure 1 represent the transmitting aperture; the coordinates are represented by $(x_t, y_t, z_0)$, and the red dots represent the receiving aperture, and the coordinates are represented by $(x_r, y_r, z_0)$. The imaging area is represented by the letter $V$. By performing a Fourier transform on the time-domain echo signal $s_{T-R}(t)$ derived from Equation (4), echo signals in the frequency domain were achieved. The frequency domain echo signal becomes a function of the position of the transmitting antenna $(x_t, y_t, z_0)$, the position of the receiving antenna $(x_r, y_r, z_0)$, and the frequency $k$. The echoes scattered from each acquisition location $s(x_t, x_r, y_t, y_r, z_0, k)$ can be represented as

$$s(x_t, x_r, y_t, y_r, z_0, k) = \iiint_V O(x', y', z') \times \frac{1}{4\pi R_t R_r} \exp[-jk(R_t + R_r)]dx'dy'dz' \quad (5)$$

where

$$R_t = \sqrt{(x_t - x')^2 + (y_t - y')^2 + (z_0 - z')^2} \quad (6)$$

$$R_r = \sqrt{(x_r - x')^2 + (y_r - y')^2 + (z_0 - z')^2} \quad (7)$$

After the completion of the Fourier transform, spatial matched filtering, stolt interpolation, wavenumber domain rearrangement, and an inverse Fourier transform, the reconstructed image of the 3-D target can be obtained.

$$O(x', y', z') = IFT_{3D}\left\{\{FT_{4D}[s(x_t, x_r, y_t, y_r, z_0, k)]\exp(-jk_z z_0)\}_{\text{stolt interpolation+rearrange}}\right\} \quad (8)$$

### 2.3. Element Spacing and Spatial Resolution

To avoid aliasing effects, the phase difference between adjacent transmitting and receiving elements usually must be less than $\pi$ rad [25]. Using $R_{t1}$ and $R_{t2}$ to represent the phase distance corresponding to a group of adjacent transmitting and receiving elements, it should satisfy

$$k|R_{t1} - R_{t2}| \leq \pi \quad (9)$$

Thus, the spacing between transmitting and receiving elements should meet the following requirements

$$d_t \leq \lambda_{\min} \frac{\sqrt{(L_t + D_x)^2/4 + r_0^2}}{L_t + D_x} \tag{10}$$

$$d_r \leq \lambda_{\min} \frac{\sqrt{(L_r + D_x)^2/4 + r_0^2}}{L_r + D_x} \tag{11}$$

where $L_t$ and $L_r$ are the lengths of the transmitter and receiver arrays, respectively, $\lambda_{\min}$ is the minimum wavelength in the frequency band, $D_x$ is the azimuthal width of the imaging region, and $r_0$ is the minimum distance from the array to the imaging region.

The range resolution $\delta_r$ is

$$\delta_r = \frac{c}{2B} \tag{12}$$

The azimuth resolution of the array direction $\delta_y$ is

$$\delta_y \approx 0.886 \times \frac{\lambda_c r}{L_t + L_r} \tag{13}$$

where $\lambda_c$ represents the wavelength corresponding to the center frequency and $r$ represents the actual distance from the array to the targets.

It can be concluded that azimuth resolution is mainly related to radar signal frequency, array length, and target distance, while range resolution is mainly determined by system bandwidth.

*2.4. MIMO Array Design and Implementation*

Virtual arrays are a commonly used equivalent form of MIMO array. By using the equivalent phase center (EPC) method, a pair of transmitting and receiving elements in a MIMO array can be equivalent to a single transmitting and receiving virtual element [26]. A MIMO array with $N_t$ transmitters and $N_r$ receivers is equivalent to $N_t \times N_r$ phase centers, according to the center position of the transceiver combination. This array is called a virtual equivalent array, which assumes that the echo data is collected by a pair of transmitting and receiving antennas at this phase center.

In order to obtain more virtual array elements with as few transceiver units as possible, and considering the consistency of the system, the system is implemented by forming an array antenna with multiple antenna periodic structures in order to achieve effective sparsity efficiency, and reduce the complexity and cost of the system.

The imaging system consisted of ten 70~80 GHz imaging transceiver array modules. The length of each module was 96 mm, where the length of the array was 90 mm, and the length of the array from the left to the right edges of the module was 3 mm. Antenna arrays for each module included $N_t = 4$ transmitters and $N_r = 16$ receivers. The spacing between the transmitters was 9 mm, and it was divided into two sub arrays, symmetrically arranged at both sides of the array, and the spacing between the receivers was 6 mm, evenly distributed throughout the entire array. The distribution diagram of a single module array and the equivalent phase center (EPC) is shown in Figure 2, and the spacing of the resulting virtual array elements is 1.5 mm.

According to the working mode of this system, all modules, except the first and tenth modules, transmit signals when working via the two transmitting units that come with the module, and a total of 32 receiving units of two neighboring modules work simultaneously. For example, when the two transmitting units on module 3 transmit signals, the echo signals are received by the 32 receiving units of modules 3 and 4. Such a working mode ensures the uniform distribution of equivalent phase centers, and, through screening, duplicate EPCs are removed to reduce the hardware costs and data computation. The entire antenna array consisted of 22 transmitting units and 160 receiving units. The combination of transmitting and receiving units was controlled by electrical signals, resulting in a total

of 640 equivalent phase centers. The consecutive and non-repetitive 633 of them were selected out for performing echo signal processing and final imaging.

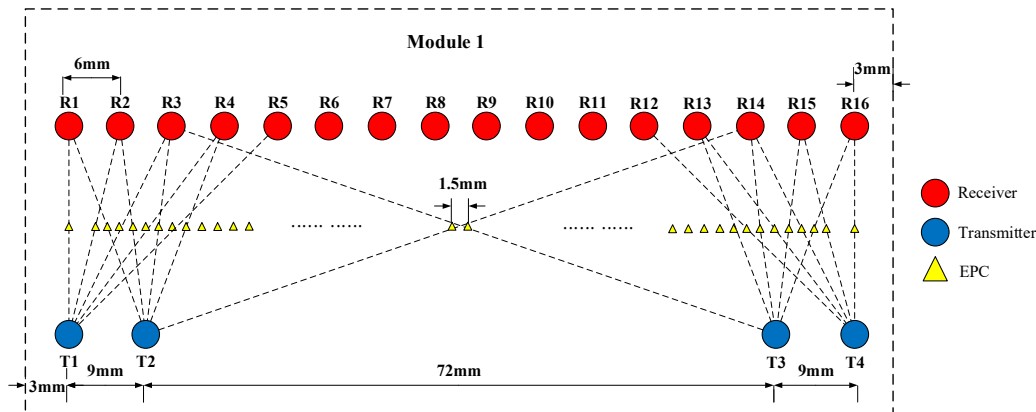

**Figure 2.** Schematic diagram of a single module array and EPC distribution.

The system controlled the work of the transceiver units in the direction of the array (electrical scanning dimension) through the program, and formed a two-dimensional aperture via mechanical scanning in a direction perpendicular to the array, which ultimately achieved the three-dimensional imaging of the tested target. The step length and elapsed length of mechanical scanning were set according to the size of the measured object, so that the antenna aperture covered the entire imaging area. The schematic diagram of the system working principle is shown in Figure 3.

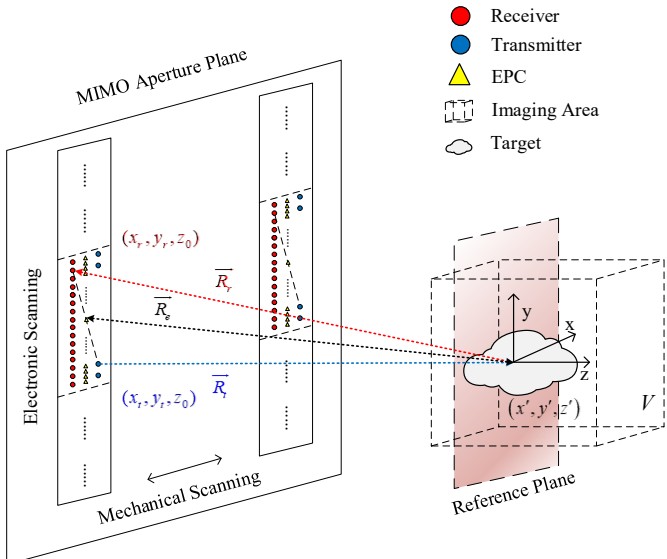

**Figure 3.** Schematic diagram of the working principle of millimeter wave MIMO imaging system.

### 2.5. Point Spread Function Simulations

In order to determine the actual spatial resolution of the imaging algorithm, a point spread function (PSF) is generally used as an analytical tool [27]. In practical imaging, the spatial extension of the response of a point target located at a certain distance directly in front of the array is used as the PSF, and the −3 dB width of the main beam of the PSF in a given dimension is generally used as the actual resolution of the imaging system in this dimension.

To verify the imaging capability of the designed array, simulations were carried out on a computer using RMA. The simulation was carried out in MATLAB R2021a on a PC with Intel® Core™ i7-9700 CPU @3.00 GHz and 16 GB RAM, and a 64-bit operating system, Microsoft Windows 10.

As can be seen from the previous text, the transmit signals were emitted by two adjacent transmitter units and received by the corresponding 32 receiver units, so only the 2-transmitter, 32-receiver array was utilized for the simulation verification. The spacing of the transmitting units was 9 mm, the spacing of the receiving units was 6 mm, and the total length of the linear array was 0.186 m. The one-dimensional line array was translated in order to simulate the mechanical scanning in the real application scenario, thus forming a two-dimensional aperture. The step length of the array translation was 1.5 mm, which is equal to the spacing of the virtual array elements, and the number of translation points was 60.

The frequency range of the transmitted signal was 70~80 GHz, and 41 points were sampled in steps of 250 MHz.

The imaging simulation was performed on a single-point target at 0.3 m from the array and a 2-D points-matrix target with a distance interval of 2 cm at 0.3 m. The schematic diagram of the simulation scene is shown in Figure 4, the obtained 2-D imaging results are shown in Figure 5, and the echo intensity curve results are shown in Figure 6.

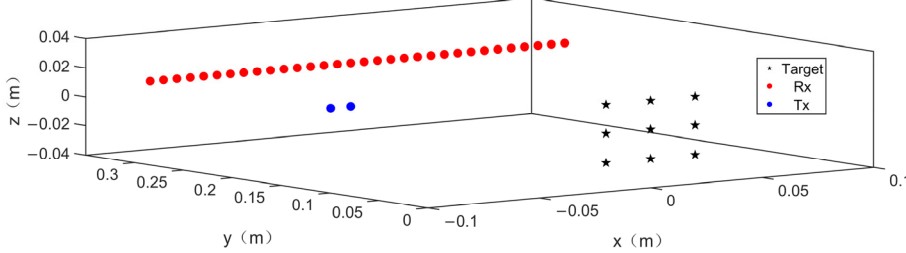

**Figure 4.** Schematic diagram of point-target imaging simulation scene.

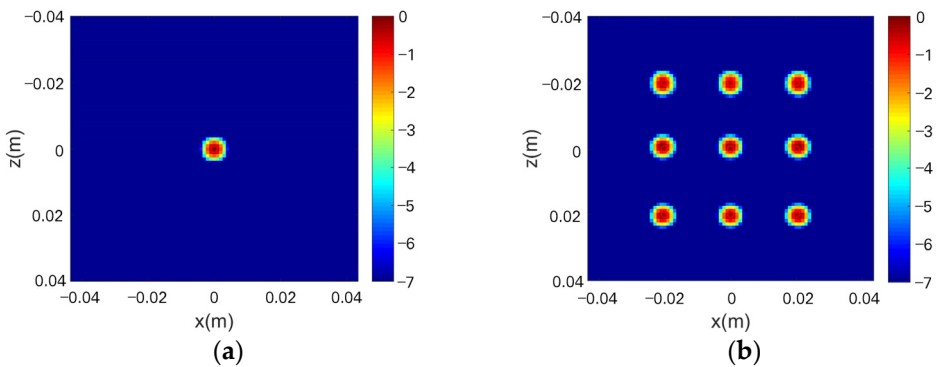

**Figure 5.** (**a**) Simulation results of point targets at 0.3 m; (**b**) simulation results of two-dimensional points-matrix target at 0.3 m.

According to the system working mode, and the way the transmitting and receiving units generate EPCs, the theoretical azimuthal resolution can be calculated. Substitute $\lambda_c = 4\,\text{mm}$, $r = 0.3\,\text{m}$, $L_t = 9\,\text{mm}$, and $L_t = 186\,\text{mm}$ into Equation (13)

$$\delta_x \approx 0.886 \times \frac{\lambda_c r}{L_{xt} + L_{xr}} = 0.886 \times 0.004 \times \frac{0.3}{0.186 + 0.009} = 0.0055\,\text{m} \tag{14}$$

Use the method of taking 3 dB width of the echo curve to calculate the resolution. It can be calculated from Figure 6a; the azimuthal resolution $\delta_{x0}$ and resolution of mechanical motion direction $\delta_{z0}$ are 5.541 mm and 5.123 mm, respectively. When the target is points matrix, the resolution can be calculated from Figure 6c,d. $\overline{\delta_x}$ and $\overline{\delta_z}$ are 5.55 mm and 5.235 mm, respectively. The azimuthal resolutions of both single point and points matrix remains consistent with the theoretical value.

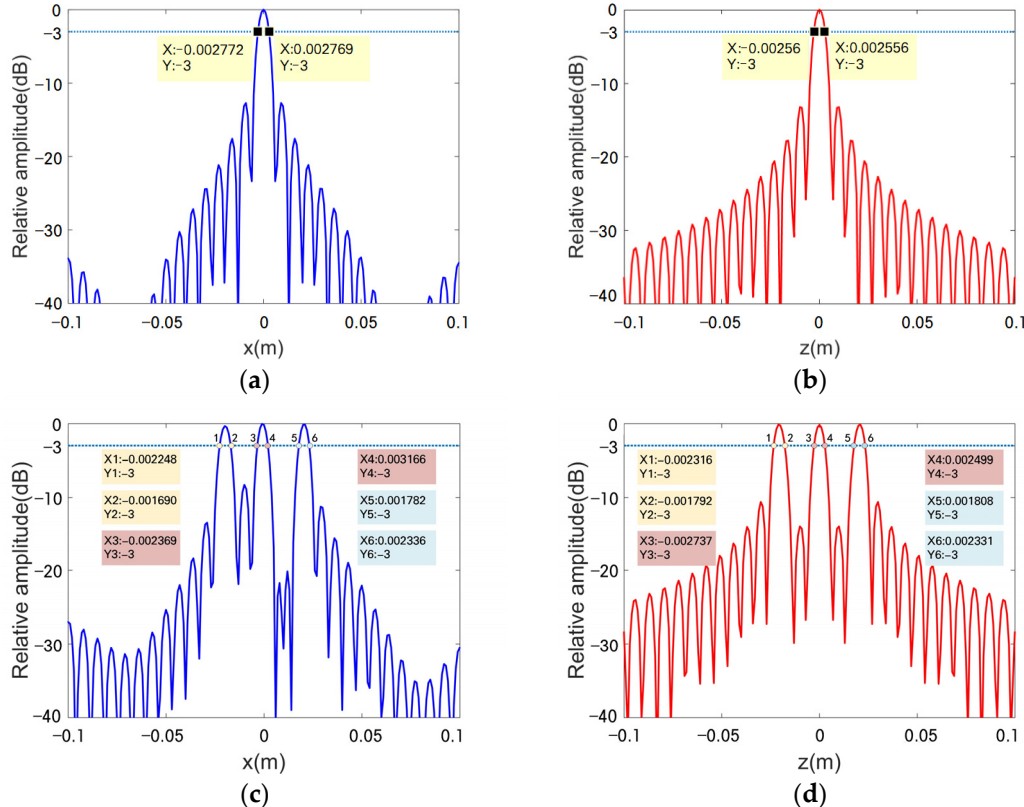

**Figure 6.** Single-point target and two-dimensional point-target matrix echo curves at 0.3 m. (**a**) Azimuth direction echo curve of single-point target; (**b**) mechanical scanning direction echo curve of single-point target; (**c**) azimuth direction echo curve of points-matrix target; (**d**) mechanical scanning direction echo curve of points-matrix target.

## 3. Radar System Description

A millimeter wave MIMO radar system was designed and built. Additionally, we have completed a system structure and parameter design, system channel calibration, antenna array design, and hardware debugging. This section mainly introduces the system structure, RF chip, and channel calibration methods.

### 3.1. System Structure

The whole system can be functionally divided into three parts, which are the motion control module, the signal generation and processing module, and the imaging computer module. The structure schematic of the system is shown in Figure 7.

When the system works, the computer first gives instructions, and the FPGA module sends the trigger signal to a coherent signal source with two channels and adjustable pulse width. After passing through a phase-locked loop circuit, the output frequency is 17.5~20 GHz and 17.5~20 GHz + $\Delta f$ FMCW signals, which are respectively sent to the transmission signal power division network and the reference signal power division network, where $\Delta f = 25$ MHz is the differential frequency signal. The power division network mainly consists of frequency multipliers and power dividers, which are used to amplify the signal to the 70~80 GHz operating frequency required by the system and distribute the signal power to ten RF transceiver modules. The switch selection matrix is directly controlled by the FPGA to determine the antenna unit that participates in the signal transceiver each time. The signal is transmitted to the corresponding antennas (the red symbols represent receiving antennas and blue symbols represent transmitting antennas) and reference channels through the power division network. The received signal passes through a mixer and a bandpass filter, and the obtained intermediate frequency (IF) signal is sent to the ADC module. The center frequency of the IF signal is 100 MHz. The ADC

module simultaneously receives the reference clock signal from the signal source and the control signal from the FPGA, and performs analog-to-digital conversion of data. The converted digital signal and the reference clock signal are transferred to the FPGA and transmitted to the imaging computer via Ethernet, thus completing the data acquisition of a mechanical point. After collecting the echo data of a point, the computer sends instructions to the motion control system, which is driven by a motor to mechanically move the transmitting and receiving antenna. The data collection for the next point is carried out, and the above process is repeated until all points in the imaging area are scanned. The obtained data is finally subjected to subsequent imaging processing.

The photograph of the imaging system is shown in Figure 8.

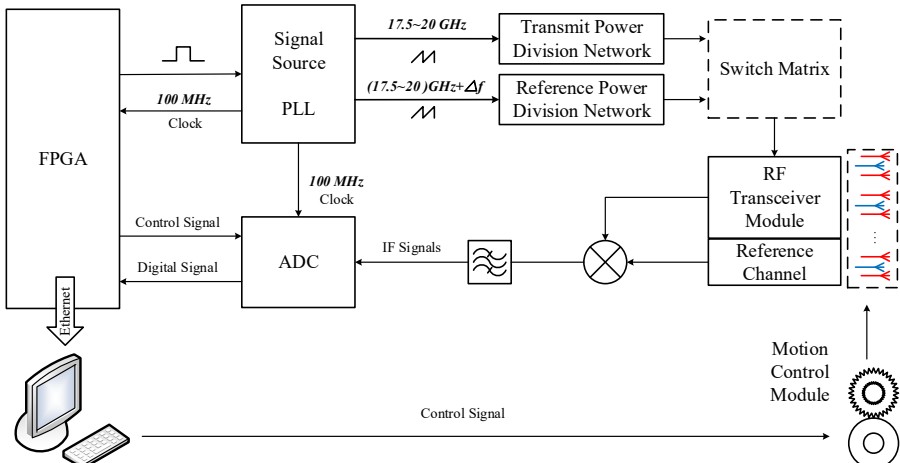

**Figure 7.** Schematic diagram of radar system structure.

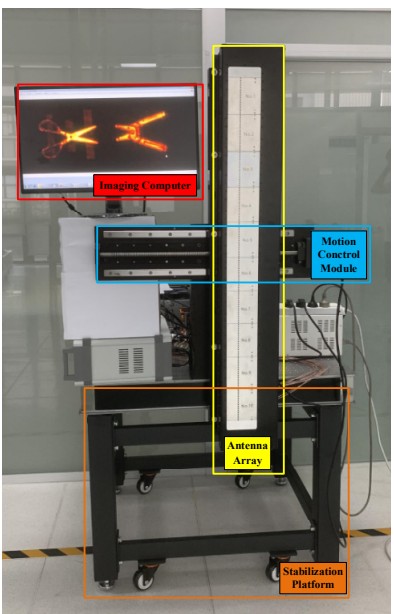

**Figure 8.** Photograph of the imaging system.

### 3.2. RF Chip

The system adopts a close-range millimeter wave imaging RF module, which uses a millimeter wave chip and a sparse linear array structure, high-performance PCB substrates, and an aluminum alloy cavity. The module is internally integrated with a four-fold frequency circuit. It transmits signals and receives a local oscillator frequency of 1/4 of the working frequency, which can reduce the technical requirements for interconnecting cables. The parallel reception operation reduces data acquisition time, and the module can

be equipped with different intermediate frequency switches to balance speed and cost. The structure of the module is shown in Figure 9.

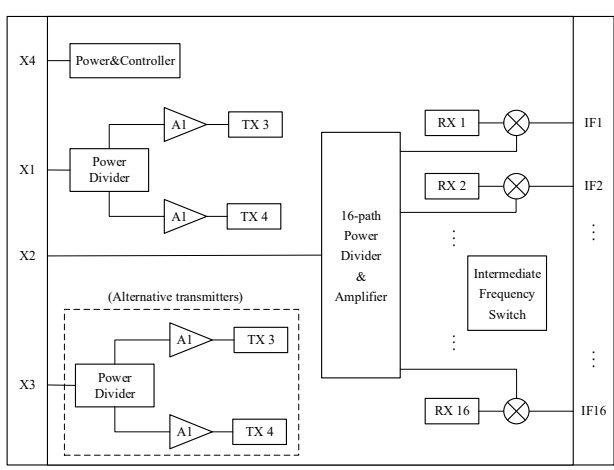

**Figure 9.** Schematic diagram of the imaging module structure.

### 3.3. Channel Calibration

A MIMO array consists of multiple transmitter (Tx) and receiver (Rx) units, with each antenna connected to a channel. Each channel in the multistatic array causes a time delay and magnitude change to the transmitted, as well as the received, signals. The main systematic sources of errors include the intrinsic characteristics of each transmitting and receiving channel, the cross-coupling between the transmitting and receiving channels, the mutual coupling between the various transmitting channels, and the mutual coupling between the various receiving channels. The channels are designed to behave linearly over the signal power levels of interest. Therefore, a linear model can be used to correct the systematic errors caused by the channels [28].

Traditional channel calibration techniques require setting up an extra reference channel outside the receiving channel in order to calibrate the actual receiving channel through the reference signal of the reference channel, which has defects such as the inability to calibrate in real time, inconvenience in calibration, and systematical errors.

The calibration method in this paper involves two aspects. Firstly, the 'direct wave' is separated from the original echo signal. The amplitude, phase and time delay of the direct wave, and the original received signal are completely consistent with those of the original received signal, which can meet the calibration requirements of channel consistency. Secondly, the total reflection of a metal plate is adopted for calibration. The metal plates are placed at different distances in front of the radar in order to obtain the total reflection echoes at the distance. The total reflection echoes at each distance are stored and can be used for the subsequent error correction. Both methods can be performed simultaneously without conflicting with each other.

The steps to obtain the direct wave are as follows:

1.  Perform a fast Fourier transform on the received signal to obtain the transformed received signal;
2.  Select the direct wave in the frequency domain from the transformed received signal through a preset frequency domain window;
3.  Perform an inverse fast Fourier transform on the direct wave in the frequency domain to obtain the direct wave.

A direct wave is the signal waveform that is directly coupled (fed) from the transmitter to the receiver without target reflection. The signal waveform transmitted from the transmitting channel through the feed-through path via spatial coupling, couplers, cables, dielectrics, etc., either individually or in combination, is weak and has a negligible effect on

the direct wave, which does not affect the feature of the direct wave. Then the direct wave can be easily separated and utilized in order to calibrate the imaging system.

The imaging device system shown in Figure 10 comprises various components like a reference clock, trigger controller, frequency scanning controller, linear frequency modulation sources, modules for selection and calibration, RF transceiver, mixer, filter, ADC, delay compensator, calibration plane, database, and imaging processor.

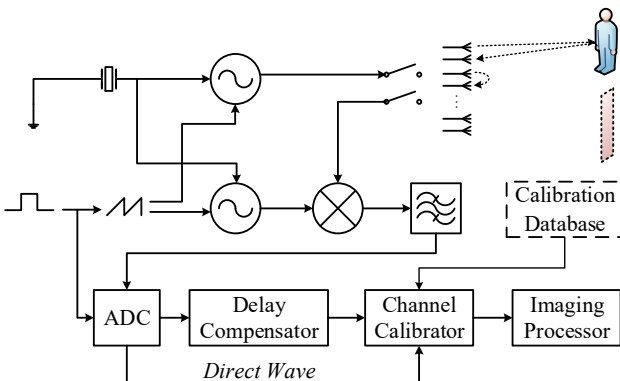

**Figure 10.** Schematic diagram of calibration equipment structure.

Before operation, a calibration plate replaces the target, gathering received and reference signals to compute errors, and generate calibration data and store it in the database.

During operation, the reference clock stabilizes the imaging device's frequency. The trigger controller syncs the frequency scanner with the receiving ADC, while the frequency scanning controller manages synchronous scanning of modulation sources. The difference between frequencies forms an intermediate frequency, which undergoes signal transmission and mixing in various stages for digital conversion.

A portion of the signal goes through delay processing and calibration, based on pre-processed data, achieving calibration of the channel and producing calibrated data sent to the imaging processor for imaging completion.

Since the direct wave passes through all the paths of the channel, reflecting the amplitude, phase, and delay information of the channel, correction parameters can be directly calculated based on the direct wave. The calibration of the amplitude, phase, and delay of the channel can be accomplished by the complex operation of the direct wave and the input signal. The self-calibration method utilizes the receiving channel exclusively, avoiding extra hardware. The software separates and calibrates the direct wave, streamlining the process and enabling real-time calibration without electrical parameter drift issues.

In addition, the imaging system can also use mechanical scanning, electrical scanning, or a combination of the two scanning modes, in order to collect signals from different positions. Based on the amplitude, phase, and time delay information between the received signal and the transmitted signal, the shape, position, material, velocity, and other information of the target are obtained through digital processing, and a target image is generated.

## 4. Measurement Results

In order to validate the imaging capability of the system, actual target detection and result analyses were performed in the laboratory.

The system adopted frequency modulated continuous wave, the carrier frequency $f_c$ was 75 GHz, band $B$ was 10 GHz, sampling frequency $F_s$ was 10 MHz, and the pulse duration $T$ was 12.8 μs.

In accordance with the proposed calibration method, the system first to collected data from the air, that is, with no targets placed in the imaging area of the system. Afterwards, a steel plate was placed in front of the radar to collect data. The size of the steel plate was 1.5 m × 1 m, larger than the majority of the targets' measured sizes, and

covered the entire imaging area. The iron plate was placed in front of the radar several times at different distances, such as 10 cm, 20 cm, 30 cm, and 40 cm, and the distance setting included the entire imaging space. The data were collected and processed for subsequent channel error correction.

The speed of mechanical scanning was controlled by a motor. The scanning speed needs to balance the quality of the collected data and the duration of the collection process, and was finally determined to be 0.3 m/s. According to the size of the imaging target, the number of mechanical scanning points was set, and the step size of the mechanical scanning was 1.5 mm.

Experiments were carried out on metal spherical shells, a resolution test board, and a human body model, respectively.

Three metal spherical shells with diameters of 1 cm, 2 cm, and 3 cm were placed 30 cm directly in front of the array in order to simulate an ideal point target, and were scanned and imaged by the system. The number of mechanical scanning points was 100, i.e., the total scanning width was 15 cm. The photograph of the experiment scene and the imaging results are shown in Figure 11. The echo intensity profiles of the target along the array direction and the mechanical scanning direction are shown in Figure 12.

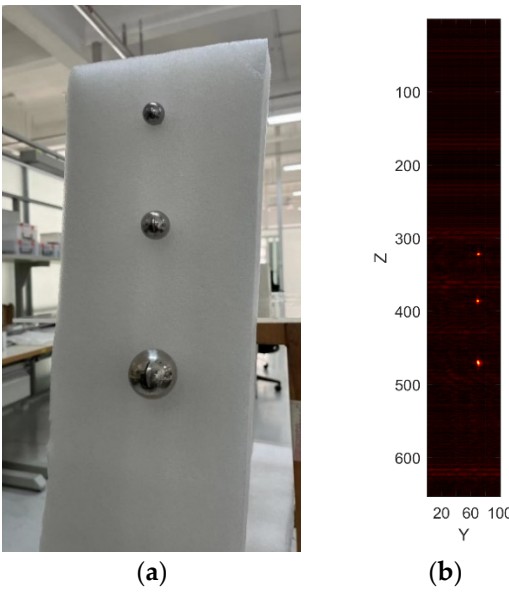

(**a**)　(**b**)

**Figure 11.** (**a**) Photo of the experimental scene, including three metal spherical shells at 0.3 m; (**b**) imaging results of three metal shells.

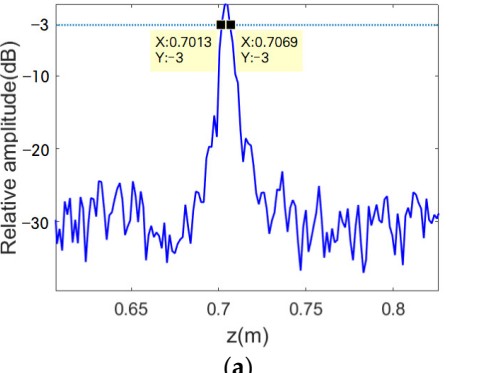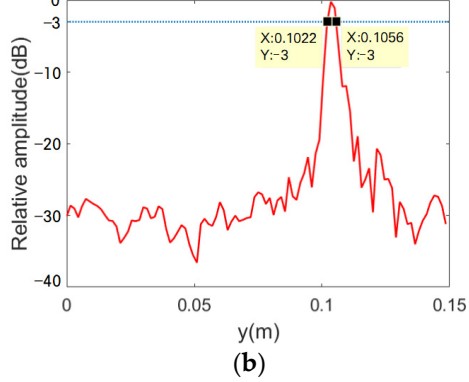

(**a**)　(**b**)

**Figure 12.** Echo curves of three metal spherical shells at 0.3 m. (**a**) Azimuth direction echo curve; (**b**) mechanical scanning direction echo curve.

Three metal spherical shells were clearly imaged by the system. The data labels in Figure 12 were used to calculate a 3 dB width. The azimuthal resolution reached 5.6 mm, which is very close to the theoretical calculated value of 5.5 mm. The resolution of mechanical scanning direction reached 3.4 mm.

The resolution test board consists of a series of rectangular apertures etched on a metal plate, with a maximum width of 4.5 mm and a minimum width of 0.8 mm, and was used to verify the azimuthal resolution of the imaging system. The resolution test board was placed in front of the imaging system at a distance of 0.25 m, and the system was used to scan and image. The optical photographs and the imaging results before and after calibration are shown in Figure 13. From the imaging results, it can be seen that the calibration method improves imaging quality. After calibration, the stripes with a spacing of 2.9 mm on the resolution plate can be clearly recognized, and the 2.5 mm stripes can also be distinguished from each other.

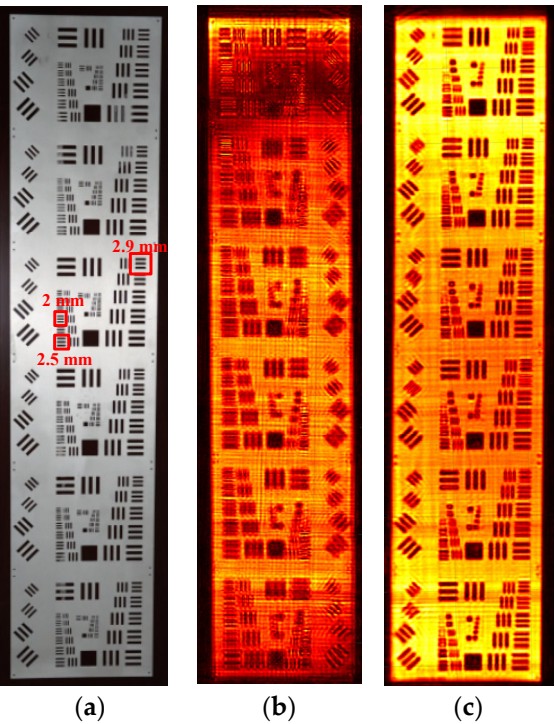

|  (a)  |  (b)  |  (c)  |

**Figure 13.** (**a**) Optical picture of the resolution board; (**b**) imaging results before calibration; (**c**) imaging results after calibration.

In order to simulate real security checks, imaging experiments were conducted on a human body model carrying metal dangerous articles. The model was made mainly of silicone and plastic, and the clothes on the model were made of cotton and polyester fibers. There was a metal pistol model hanging on the model's chest, which was 0.25 m away from the antenna array. A dagger was tied to the left arm, the overall length was about 10 cm, and was 0.4 m away from the antenna array. The optical photos of the pistol model and dagger are shown in Figure 14. The pistol model and the dagger hidden underneath the clothes, which were located at different distances, were clearly imaged by the system. The photos of the experimental scene and imaging results are shown in Figure 15.

In addition, stability testing was conducted on the system. The decrease in image quality was negligible when running continuously for more than 12 h.

The above experimental results indicate that the constructed system has excellent imaging resolution, including range resolution and azimuth resolution. The system has a certain penetration ability and can detect dangerous metal objects hidden under clothing. The system meets the resolution requirements for targets in the imaging area, and is

effectively compatible with fast imaging algorithms in the wavenumber domain to ensure imaging speed, proving the effectiveness of the imaging system.

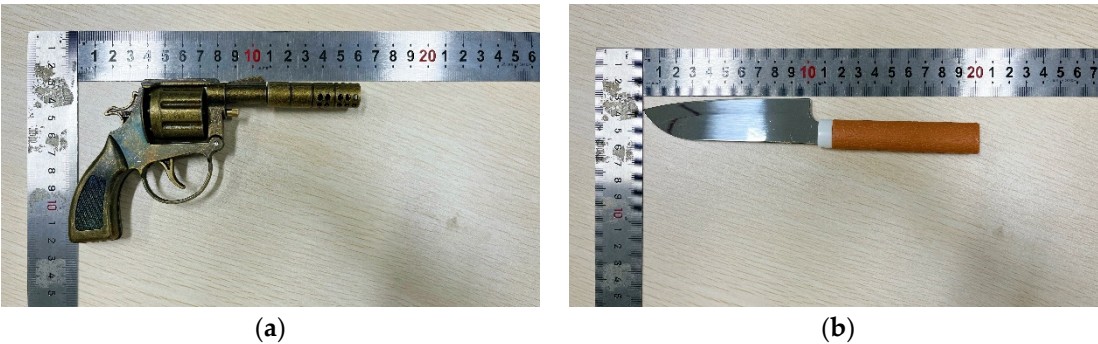

(**a**)　　　　　　　　　　　　　　　　(**b**)

**Figure 14.** (**a**) Optical photo of the pistol model; (**b**) optical photo of the dagger.

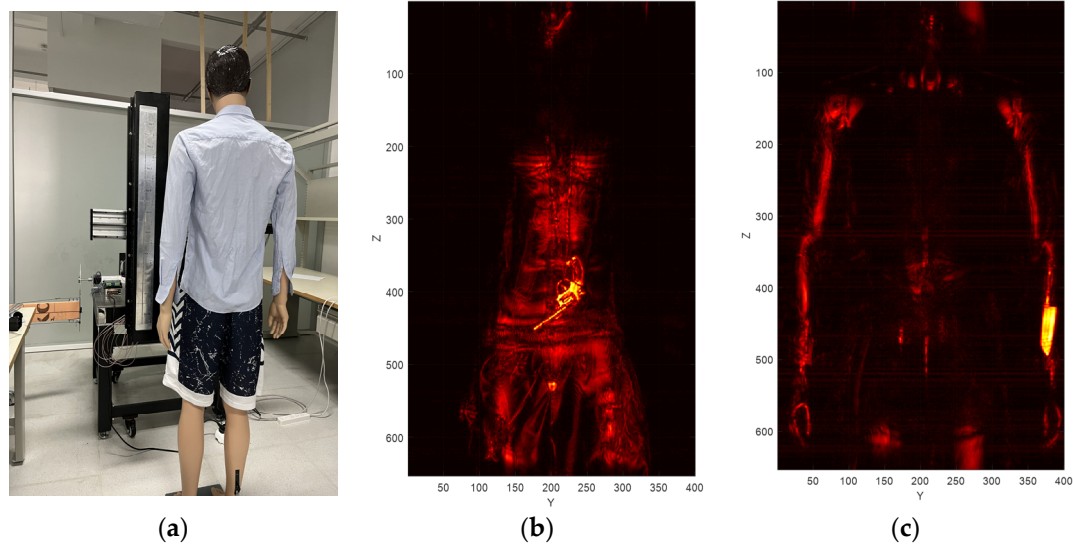

(**a**)　　　　　　　　　　(**b**)　　　　　　　　　　(**c**)

**Figure 15.** (**a**) Photo of the experimental scene, including the imaging system and human models containing dangerous articles hidden under clothing; (**b**) imaging results of the pistol model; (**c**) imaging results of the dagger.

## 5. Conclusions

In this paper, the design of a millimeter wave MIMO imaging system in the FMCW regime is proposed, and experiments were conducted based on the designed system to demonstrate the imaging resolution. An operating frequency band of 70~80 GHz and a 10 GHz bandwidth guarantee the accurate azimuthal and distance resolution of the system, respectively. The channel calibration method adopted in the system suppresses the intrinsic error of the system channels, and there is no need to set up extra physical channels, reducing the hardware cost. The sparse MIMO array maximizes the efficiency of array element utilization. The wavenumber domain imaging algorithm was applied to the system and achieved excellent imaging results.

**Author Contributions:** Conceptualization, W.S.; methodology, W.S. and J.H.; software, J.H.; validation, W.S.; formal analysis, W.S.; investigation, W.S., W.Z. and J.H.; data curation, W.S., W.Z. and J.H.; writing—original draft preparation, W.S.; writing—review and editing, W.S., J.H. and Y.J.; supervision, J.H. and Y.J.; project administration, G.F.; funding acquisition, J.H. and G.F. All authors have read and agreed to the published version of the manuscript.

**Funding:** This work was supported in part by the National Natural Science Foundation of China under grant 61988102, grant 61731020, and grant 61971397; in part by the Pearl River Talent Plan under grant 2021QN02Z134; and in part by the Key Projects of the National Natural Science Foundation of China under grant 6233000092.

**Data Availability Statement:** Data are contained within the article.

**Conflicts of Interest:** The authors declare no conflicts of interest.

## Abbreviations

The following abbreviations are used in this manuscript:

| | |
|---|---|
| MIMO | multiple-input multiple-output |
| FMCW | frequency modulated continuous wave |
| 3-D | three-dimensional |
| 2-D | two-dimensional |
| SISO | single-input-single-output |
| SAR | synthetic aperture radar |
| RMA | range migration algorithm |
| EPC | equivalent phase center |
| PSF | point spread function |
| RF | radio frequency |

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
