# Peer review of "W-Band FMCW MIMO System for 3-D Imaging Based on Sparse Array"

_electronics, doi:10.3390/electronics13020369_

Round 1

Reviewer 1 Report

Comments and Suggestions for Authors

Please find enclosed my comments and suggestions.

Comments on the Quality of English Language

The article is well written.

Author Response

Thank you for your valuable comments and suggestions on our work. We have added some background information, knowledge about channel calibration, photograph of imaging system, and an explanation of experimental results. The detailed corrections are presented with an attachment.

Reviewer 2 Report

Comments and Suggestions for Authors

 The paper presents a sparse array-based frequency-modulated continuous wave (FMCW) millimeter-wave imaging system operating in the W-band (70~80 GHz). Employing Multiple-Input-Multiple-Output (MIMO) technique, it reduces transceiver units, lowering hardware costs. The system combines mechanical and electrical scanning, utilizing FMCW signals for distance extraction

Some important points have to be clarified or justified and few concerns need to be addressed by the authors for the betterment of the manuscript

1-      How does the equivalent phase center (EPC) method contribute to reducing the number of transceiver units, and what advantages does it offer in comparison to traditional MIMO array configurations?

2-      It is recommended to include the following article:-

Elhanashi, A., Gasmi, K., Begni, A., Dini, P., Zheng, Q., Saponara, S. (2023). Machine Learning Techniques for Anomaly-Based Detection System on CSE-CIC-IDS2018 Dataset. In: Berta, R., De Gloria, A. (eds) Applications in Electronics Pervading Industry, Environment and Society. ApplePies 2022. Lecture Notes in Electrical Engineering, vol 1036. Springer, Cham. https://doi.org/10.1007/978-3-031-30333-3_17

3-      Could you explain in more detail the rationale behind the working mode where all modules, except the first and tenth, transmit signals simultaneously? How does this contribute to achieving uniform distribution of equivalent phase centers?

4-      How does the system achieve a 2 mm imaging resolution, and how was this resolution validated through simulations and experiments?

5-      The article mentions channel calibration and a calibration plate. Could you provide more details on how calibration is performed and its significance in enhancing imaging quality?

6-      The article effectively addresses the hardware and cost-related challenges in existing imaging systems and proposes a comprehensive solution. However, a more detailed explanation of the imaging algorithm are required to  enhance clarity.

7-      The article should include  a brief comparison with other existing millimeter wave imaging technologies, highlighting the unique advantages of the proposed system.

8-      Please include  the potential limitations and future improvements of the proposed system?

Comments on the Quality of English Language

Further proofreading is required

Author Response

(The authors gave the same response as above.)

Round 2

Reviewer 2 Report

Comments and Suggestions for Authors

Thanks to author for implementation and enhancing the manuscript